# Effects of Fermented Bamboo Shoot Processing Waste on Growth Performance, Serum Parameters, and Gut Microbiota of Weaned Piglets

**DOI:** 10.3390/ani12202728

**Published:** 2022-10-11

**Authors:** Yuanhao Huang, Yingjie Peng, Zheng Yang, Siyu Chen, Jing Liu, Zheng Wang, Gang Wang, Shile Lan

**Affiliations:** 1College of Bioscience and Biotechnology, Hunan Agricultural University, Changsha 410128, China; 2Hunan Huajun Agricultural Technology Co., Ltd., Taojiang 413000, China

**Keywords:** fermented bamboo shoot processing waste, weaned piglet, growth performance, serum parameters, gut microbiota

## Abstract

**Simple Summary:**

Bamboo shoot processing waste is usually directly discarded, resulting in the waste of bamboo resources and environmental pollution. Fermented bamboo shoot processing waste (FBSPW) for feed processing is an important way to reuse bamboo shoot processing waste. In this study, to evaluate the feasibility of FBSPW as a feed additive for weaned piglets, we analyzed the effects of different concentrations of FBSPW added to feed on growth performance, serum parameters, and gut microbiota of weaned piglets. Our results showed that the addition of FBSPW significantly decreased the average daily feed intake, serum triglyceride content, and urea nitrogen of weaned piglets compared to the control. The cecum and cecal microbiota of weaned piglets fed the basal diet with 12% FBSPW were significantly different compared to the control. The addition of 12% FBSPW to weaned piglet feed could improve their nitrogen and lipid metabolisms and have beneficial effects on gut microbiota.

**Abstract:**

Gut microbiota (GM) plays a vital role in the nutrition and metabolism of weaned piglets. Some feed additives can be used to adjust the composition of GM to improve the health of weaned piglets. In this study, we investigated the effects of adding fermented bamboo shoot processing waste (FBSPW) to diet on growth performance, serum parameters, and GM of weaned piglets. Seventy-two piglets were divided into four groups and were fed diets containing 0% (control), 4% (group A), 8% (group B), and 12% (group C) FBSPW for 50 days. We found that the addition of FBSPW significantly decreased the average daily feed intake, serum triglyceride content, and urea nitrogen of weaned piglets compared to the control. The cecum and cecal microbiota of weaned piglets fed the basal diet with 12% FBSPW were significantly different compared to the control. A basal diet with 12% FBSPW significantly reduced the taxon feature number, and the relative abundance of Tenericutes in the cecum and cecal microbiota of weaned piglets compared with the control. The addition of 12% FBSPW to weaned piglet feed could improve their nitrogen and lipid metabolisms and have beneficial effects on GM.

## 1. Introduction

Bamboo shoots are a kind of common natural vegetable food in southern China. They are loved by Chinese people for their high protein, low fat, low sugar, and high fiber [1]. Nearly 60% of bamboo shoots can only be processed into dried and canned bamboo shoots every year because of the difficulty of keeping them fresh [2], resulting in a large amount of bamboo shoot processing waste (BSPW), such as bamboo shoot shell, shoot head, and shoot residue. Presently, BSPW is mainly directly discarded, leading to the waste of bamboo resources and environmental pollution [3,4]. Moreover, BSPW is difficult to use.

The protein, sugar, and cellulose contents of BSPW are 131.4 g/kg, 115.3 g/kg, and 235.4 g/kg, respectively [5]. Compared with that of other coarse fiber animal feeds such as wheat and rice straws, the crude protein content of BSPW is higher and can produce better economic benefits as a feed additive [6]. As a feed additive, BSPW significantly reduced the blood lipid content and increased the weight of mice [7]. Therefore, using BSPW as animal feed will not only prevent the environmental pollution caused by the direct disposal of BSPW but also optimize the resource utilization of BSPW.

Weaning is the most important process in the growth of pigs. The change of food from breast milk to solid feed during weaning is often accompanied by changes in gut microbiota and immunology, which may result in diarrhea, slow growth, and even death [8]. Antibiotics are often added to diets as feed additives in the animal breeding industry, but the long-term use of antibiotics leads to the development of antibiotic resistance [9]. Therefore, we urgently need to find a natural and healthy preparation to improve the gut microbial health of weaned piglets and alleviate the adverse effects of stress caused by weaning on piglets.

Microbial fermentation can effectively remove anti-nutritional factors from BSPW [10] and increase its palatability [11]. Fermented bamboo shoots can accelerate intestinal peristalsis, relieve constipation, and facilitate the absorption of nutrients by the intestine [12]. However, the effect of fermented BSPW (FBSPW) on the gut microbiota of weaned piglets and its correlation with piglet growth has not been reported. Considering that fermented feed additives can significantly change the gut microbiota of animals and promote animal growth [13,14,15], we hypothesized that FBSPW will significantly change the gut microbiota of weaned piglets and promote the growth of weaned piglets. To test this hypothesis, we analyzed the effects of FBSPW on growth performance, serum indices, and gut microbiota of weaned piglets through physiological and biochemical determination and high-throughput sequencing.

## 2. Materials and Methods

### 2.1. Production of Fermented Feed

BSPW was purchased from a bamboo shoot processing plant in Taojiang County (Yiyang, China). *Bacillus* sp., *Lactobacillus fermentum*, and *Lactobacillus rhamnosus* for fermentation were provided by the Microbial Preparation Room of the College of Bioscience and Biotechnology of Hunan Agricultural University. The ratio of *Bacillus* sp. to *L. fermentum* and *L. rhamnosus* in the fermentation seed solution was 1:2:2. The total biomass in the fermentation seed solution was 2.4 × 10^9^ CFU/g. BSPW was fermented in fermentation bags with a one-way breathing valve, and the inoculation amount of fermentation seed solution was 10% (V/W). The BSPW was anaerobically solid-stated fermented at 38 °C for 7 days, and then dried at 50 °C for 2 days to reduce the water content to 10%.

### 2.2. Experimental Design

Animal experiments in this study were approved by and performed in accordance with the guidelines of the Biomedical Research Ethics Committee of Hunan Agricultural University (approval number: Lunshenke 2022 No. 74). The experiment was conducted on a farm at Taojiang County (Yiyang, China). Seventy-two Duroc × Yorkshire × Landrace hybrid piglets of 22.14 ± 3.52 kg were randomly divided into three treatment groups and a control group according to their body weight and sex, with six replicates group and each replicate group containing three piglets. The experimental groups contained A (piglets fed the basal diet with 4% FBSPW), B (piglets fed the basal diet with 8% FBSPW, C (piglets fed the basal diet with 12% FBSPW), and control (piglets fed the basal diet without FBSPW; Table 1). The piglets were weaned at 25 days of age and purchased by the breeding company at 31 days of age. The piglets were pre-fed with the basal diet for seven days to alleviate the stress response of the piglets due to changes in the living environment and transportation process and then fed the experimental feeds for 50 days according to the experimental design. Weaned piglets were fed ad libitum during the experiment, allowed to drink plenty of water, and immunized according to normal immunization procedures, i.e., the piglets were vaccinated with classical swine fever vaccine at 1–2 h before colostrum, attenuated pseudorabies vaccine at age of 3 d, inactivated asthmatic vaccine at age of 10 d, swine streptococcosis vaccine at age of 17 d, attenuated paratyphus vaccine for piglets at age of 25 d, inactivated asthmatic vaccine at age of 30 d, and classical swine fever vaccine at age of 60 d.

### 2.3. Determination of Growth Performance

The piglets were fed twice a day at 7:00 a.m. and 7:00 p.m., respectively. The feed and the remaining feed were weighed every day during the experiment to calculate the average daily feed intake (ADFI). The piglets with empty stomachs were weighed before feeding on the first and last morning of the experiment. Average daily gain (ADG) and feed/gain ratio (F/G) were calculated from the data collected in the experiment.

### 2.4. Determination of Serum Parameters

At the end of the experiment, three piglets with an age of 89 d were randomly selected from each treatment, and blood was collected from the anterior vena cava using blood collection tubes with heparin sodium. After they were naturally stratified, the blood samples were centrifuged at 4000 r/min for 20 min at 5 °C, and then, the upper serum was collected. The serum triglyceride, total cholesterol (TC), high-density lipoprotein (HDL), low-density lipoprotein (LDL), urea nitrogen, glucose, total protein, and albumin were measured using kits (Nanjing Jiancheng Bioengineering Institute, Nanjing, China).

### 2.5. Gut Microbiota Composition Analysis

After blood collection, the piglets were euthanized by an overdose of anesthetics according to the national standard GB/T 39760-2021 of the People’s Republic of China. After dissection, the intestinal tissue was taken out immediately, and each intestinal segment was ligated. The contents in the colon and cecum were collected and put into polypropylene centrifuge tubes, then transferred into liquid nitrogen for quick freezing, and subsequently transferred to a −80 °C refrigerator for storage for DNA extraction and microbiota composition analysis.

Microbial DNA was extracted using NucleoSpin^®^96 soil DNA extraction kits (Macherey-Nagel, Düren, Germany). The concentration and quality of DNA were detected using a synergy HTX microplate reader (GeneCompang Limited, Hong Kong, China) and 1.8% agarose gel electrophoresis, respectively. The hypervariable V3-V4 region of 16S rDNA was amplified using primers 338F (5′-ACTCCTACGGGAGGCAGCA-3′) and 806R (5′-GGACTACHVGGGTWTCTAAT-3′) and sequenced using a Sequel II sequencer (Pacbio, Silicon Valley, CA, USA) at Biomarker Technologies (Beijing, China) according to a previously described protocol [16].

The reads were merged through FLASH v1.2.11 with a minimum overlap length of 10 bp and a maximum mismatch ratio of 0.2 in the overlap area to obtain the tags. The tags with lengths less than 75% of the average length of tags were filtered using Trimmomatic version 0.33, and the chimera sequences were removed using UCHIME version 8.1 to obtain high-quality tags for further analysis. The high-quality tags were clustered to taxon features at 97% sequence similarity using USEARCH version 10.0 [17]. The taxon features were taxon annotated based on the Silva release 128 database (http://www.arb-silva.de, accessed on 1 January 2022). The taxon feature number and ACE index are commonly used to indicate taxon richness in a microbiota. Phylogenetic diversity (PD), Shannon, and Simpson indices are commonly used to indicate the bio-diversity of a microbiota. And Good’s coverage is used to indicate the proportion of taxa in the microbiota covered by analyzing [18]. These alpha-diversity indices of the samples were calculated using Mothur v.1.30. Unweighted binary Jaccard algorithm was used for principal coordinates analysis (PCoA) through QIIME2.0 to compare the similarity of the microbiota compositions of samples.

### 2.6. Data Analysis

The experimental results are expressed as mean ± standard deviation. Shapiro-Wilk normality test and Bartlett test of homogeneity of variances were conducted using R 4.0.3 [19]. When the data were normally distributed and the variances were homogeneous, one-way ANOVA was conducted using R 4.0.3. Duncan’s test was used for multiple comparisons of results with a significant difference. Otherwise, the Kruskal-Wallis rank sum test with Dunn post hoc tests was conducted using R 4.0.3 with the FSA package. Boxplots and heatmap were drawn using R 4.0.3. Redundancy analysis (RDA) and Pearson product-moment correlation coefficient analysis were conducted using R 4.0.3 with vegan and psych, reshape2, and corrplot packages, respectively. Non-parametric multivariate analysis of variance (PERMANOVA) [20] was conducted using the vegan package in R 4.0.3. Differences with *p* < 0.05 were considered significant.

## 3. Results

### 3.1. Effects of FBSPW on Growth Performances of Weaned Piglets

The ADFI of weaned piglets fed with FBSPW decreased significantly compared with that of the control group (Kruskal-Wallis rank sum test, *χ*^2^ = 11.0, *p* = 0.012; Figure 1A), but ADG (Kruskal-Wallis rank sum test, *χ*^2^ = 1.667, *p* = 0.644; Figure 1B) and F/G ratio (Kruskal-Wallis rank sum test, *χ*^2^ = 1.667, *p* = 0.644; Figure 1C) did not change significantly.

### 3.2. Effects of FBSPW on Serum Parameters of Weaned Piglets

Compared with those of the control, there was no significant difference in serum glucose, TC, LDL, and total protein of weaned piglets fed fermented feeds (*p* > 0.05; Figure 2B,C,E,H and Appendix A); however, triglyceride content decreased significantly (*p* < 0.05; Figure 2D and Appendix A). The serum HDL and albumin of weaned piglets in group B were significantly higher than those in the control (*p* < 0.05; Figure 2A,G), and those in groups A and C also increased, but the differences were not significant (*p* > 0.05; Figure 2A,G and Appendix A). Serum urea nitrogen content of weaned piglets in groups B and C were significantly lower than that in the control (*p* < 0.05; Figure 2F), and it also decreased in group A but was not significant (*p* > 0.05; Figure 2F and Appendix A).

### 3.3. Effects of FBSPW on Gut Microbiota of Weaned Piglets

A total of 216,108 (9004.50 ± 260.64) sequences were obtained from 24 samples. For each sample, 366.46 ± 17.99 taxon features were obtained. The taxon feature number, ACE, and PD indices of colon microbiota were significantly higher than those of cecum microbiota (Appendix A) and caused the Good’s coverage of colon microbiota to be significantly lower than that of cecum microbiota (Appendix A). However, the Shannon and Simpson indices of colon microbiota did not show a significant difference from those of cecum microbiota (Appendix A). Moreover, the basal diet with 12% FBSPW significantly reduced the taxon feature number, ACE, and PD indices of weaned piglet cecum and cecum microbiota compared with the control, but basal diets with 4% and 8% FBSPW did not significantly change these α-diversity indices, respectively (Figure 3A–C), making the good’s coverage of the cecum and cecum microbiota of weaned piglets fed the basal diet with 12% FBSPW to be significantly higher than that of other groups (Figure 3F). However, FBSPW supplementation did not significantly change the Shannon and Simpson indices of the weaned piglet cecum and cecum microbiota (Figure 3C,D). Moreover, cecum and cecum microbiota of weaned piglets fed the basal diet with 12% FBSPW were significantly different from those of the control (PERMANOVA, *p* < 0.05), but those of other FBSPW groups were not significantly different from those of the control (PERMANOVA, *p* > 0.05; Figure 3G,H).

Except for a few OTUs, most OTUs were divided into 11 phyla, among which Firmicutes was the most dominant phylum (Figure 4A). Although a basal diet with 4% FBSPW significantly reduced the relative abundance of Firmicutes in the cecum and colon microbiota of weaned piglets compared with those with 12% FBSPW, no significant difference was detected between the FBSPW groups and control (Figure 4B,E). A basal diet with 12% FBSPW significantly reduced the relative abundance of Tenericutes in the cecum and colon microbiota of weaned piglets compared with the control (Figure 4C,F). However, the relative abundances of Tenericutes in other FBSPW groups were not significantly different from those in the control group (Figure 4C,F). Although a basal diet with 4% FBSPW significantly reduced the relative abundance of Actinobacteria in the cecum microbiota of weaned piglets compared with the control, no significant difference was detected between the other FBSPW groups and control (Figure 4D).

In genus, the gut microbiota of control and piglet samples fed basal diets with 4% and 8% FBSPW were clustered into a single group, and the piglet samples fed the basal diet with 12% FBSPW were clustered into another group (Figure 5). Moreover, cecum and colon samples of the piglets fed the basal diet with 12% FBSPW were clustered into two distinct sub-groups (Figure 5). The relative abundances of *Streptococcus*, *Roseburia*, *Terrisporobacter*, *Prevotella*, *Solobacterium*, *Holdemanella*, *Actinobacillus*, *Blautia*, *Dorea*, *Butyricicoccus*, *Ruminiclostridium*, *Eubacterium xylanophilum*, *Oscillospira*, *Alloprevotella*, *Eubacterium*, *Clostridium*, and multiple unidentified genera were significantly different among groups (Figure 5 and Appendix A). Compared with those in the control, the relative abundances of *Holdemanella*, *Butyricicoccus*, *Solobacterium*, *Clostridium*, and *Alloprevotella* in the cecum microbiota of piglets fed the basal diet with 12% FBSPW were significantly enriched, whereas that of *Oscillospira* was significantly reduced (*p* < 0.05; Appendix A); however, no significant difference was detected in the colon microbiota (*p* > 0.05; Appendix A). Moreover, the relative abundance of *Blautia* in the colon microbiota of piglets fed the basal diet with 12% FBSPW was significantly enriched (*p* < 0.05; Appendix A). Although the cecum microbiota exhibited an increasing trend, no significant difference was detected (*p* > 0.05; Appendix A). These results implied that the effect of FBSPW on the cecum microbiota of weaned piglets was more obvious than on the colon microbiota.

Although RDA results revealed that only serum glucose significantly correlated with colon microbiota (Figure 6A,C). Pearson product-moment correlation coefficient analysis showed that serum total protein was significantly positively correlated with *Sphaerochaeta*, *Candidatus Saccharimonas*, and some unidentified genera; serum triglyceride significantly positively correlated with *Roseburia* and some unidentified genera; serum urea nitrogen significantly positively correlated with two unidentified genera; F/G significantly positively correlated with *Turicibacter*, *Ruminococcus*, and an unidentified genus; serum TC significantly positively correlated with *Alloprevotella* and significantly negatively correlated with *Eubacteirum xylanophilum*; and serum glucose positively correlated with an unidentified genus and significantly negatively correlated with *Akkermansia* in cecum microbiota (Figure 6B). Serum total protein significantly positively correlated with *Sphaerochaete*, *C. Saccharimonas*, and some unidentified genera and significantly negatively correlated with *Terrisporobacter*; serum triglyceride significantly positively correlated with *Roseburia* and an unidentified genus; serum urea nitrogen significantly positively correlated with *Roseburia*, *Agathobacter*, and some unidentified genera and significantly negatively correlated with *Phascolarctobacterium*; serum HDL significantly positively correlated with *Parabacteroides* and *Sphaerochaeta* and significantly negatively correlated with *Succinivibrio*; serum LDL significantly positively correlated with an unidentified genus; F/G significantly positively correlated with *Ruminococcus* and an unidentified genus; serum albumin significantly positively correlated with *Prevotella*; serum glucose significantly positively correlated with an unidentified genus and significantly negatively correlated with *Terrisporobacter*, *Escherichia*, and an unidentified genus; and serum TC significantly positively correlated with *Escherichia* and significantly negatively correlated with *Sphaerochaeta*, *Ruminococcus*, *E. xylanophilum*, and some unidentified genera (Figure 6D).

## 4. Discussion

BSPW is a by-product of processing and is cheap and high in crude protein content [4,21,22]. Therefore, it is a potentially suitable additive in animal feed. However, it contains a lot of cellulose and anti-nutritional factors, such as phytate, saponin, tannin alkaloid, and cyanogenic glycoside [23]; hence, its direct addition to animal feed is not suitable. Microbial fermentation can effectively remove anti-nutritional factors from BSPW [10,23] and increase its palatability [1,11]. In this study, compared with those of the control, the ADG and F/G of weaned piglets in the FBSPW-treated groups did not significantly change, corroborating the results of Farnworth et al. [24]. In addition, our results corroborate the conclusion of Bakare et al. [8] that the feeding of a fiber diet to pigs did not affect growth performance. Our results also showed that the F/G of weaned piglets increased as the FBSPW in the feed increased, probably due to the increase in cellulose in the feed, which affected the digestibility of the weaned piglets. Johnston et al. [25] reported that dietary fiber interferes with the decomposition and absorption of nutrients by affecting the flow rate, transit time, and combination of digestive enzymes and nutrients in the gastrointestinal tract.

The concentrations of serum total protein, albumin, and urea nitrogen are usually regarded as indicators of protein synthesis and metabolism, which are related to the growth performance of piglets. The increases in serum total protein and albumin indicate that more protein was synthesized and absorbed. Serum urea nitrogen is a product of protein degradation in vivo; hence, the decrease in serum urea nitrogen indicates that more proteins were synthesized from amino acids [26]. It is believed that serum urea nitrogen content is an important indicator of protein quality in feed. The decrease in serum urea nitrogen content reflects the high protein quality of fermented feed, that is, the amino acids are more effectively used to synthesize tissue protein [27]. In this study, the serum urea nitrogen content was significantly reduced in the piglets fed the basal diet with 8% and 12% FBSPW compared with that of the control (Figure 2F), corroborating the results of Cho et al. [28]. Moreover, the serum albumin content of piglets fed a basal diet with 8% FBSPW significantly increased compared with that of the control. The serum albumin content of other FBSPW-treated piglets also exhibited an increasing trend, but no significant difference was detected (Figure 2A). Serum glucose content is an indicator of energy supply [29]. In this study, the serum glucose contents of FBSPW-treated piglets were not significantly different compared with those of the control, corroborating the results of Lee et al. [30]. These results implied that the addition of 8% and 12% FBSPW to feed had the potential to promote the protein synthesis and growth of weaned piglets. Our results showed that there was no significant increase between the growths of weaned piglets treated with FBSPW and those of the control, probably owing to the short experimental process.

High serum total cholesterol and triglyceride contents, which are closely related to the diet, easily cause coronary heart diseases and atherosclerosis. Dietary fiber is regarded as a natural cholesterol-lowering substance [31]. In this study, the serum total cholesterol content of weaned piglets fed FBSPW did not change significantly, but the serum triglyceride content significantly reduced, corroborating the results of Ting et al. [32]. The main function of serum LDL is to transport the cholesterol synthesized by the liver to the tissue, and the main function of serum HDL is to transport the excess cholesterol from the tissue back to the liver for degradation [33]. In this study, compared with that of the control, the serum LDL content of weaned piglets treated with FBSPW did not change significantly, and the serum HDL content increased, especially in the piglets treated with 8% FBSPW, corroborating the results of Akanbi and Agarry [34]. These results indicate that the addition of FBSPW to the feed of weaned piglets improved the protein and lipid metabolisms of piglets, and this may be related to the prebiotics contained in the FBSPW because prebiotics can improve the protein and lipid metabolism functions of pigs [35,36].

Probiotics play a positive role in improving growth performance, gut microbiota [37], and immunity [38]. The use of microbial fermented feed is part of probiotic treatment. However, our result showed that the addition of FBSPW to feed did not significantly improve the growth performance of weaned piglets (Figure 1), but the gut microbiota of weaned piglets significantly changed when 12% FBSPW was added to the feed (Figure 3G,H and Figure 5). GM participates in host digestion and absorption of nutrients, and their metabolites also participate in the metabolism of weaned piglets [39]. Therefore, the changes in GM composition affect the health status of piglets. The composition and diversity of GM can be analyzed through high-throughput sequencing of 16S rDNA [40,41,42,43,44,45]. In this study, the alpha-diversity indices of groups A and B had no significant difference compared with those of the control. The taxon feature number and ACE index of group C significantly reduced compared with that of the control. These results indicate that the addition of less than 8% FBSPW to feed did not influence the GM alpha diversity of weaned piglets, but the addition of 12% FBSPW to feed significantly reduced the GM alpha diversity. Moreover, our results showed that the cecum and cecum microbiota of weaned piglets fed the basal diet with 12% FBSPW were significantly different compared with those of the control. It should be noted that the changes in species abundance in groups A, B, and C were different, showing that the effect of PBSPW on GM structure depended on the amount added to the feed. This may be due to the increase in dietary fiber and cellulose intake by piglets, which increased with FBSPW addition, consequently affecting the composition of GM, corroborating the results of Tap et al. [46]. Our results showed that at the phylum level, the sequence contents of Firmicutes and Bacteroidetes in the colon and cecum accounted for more than 60% of the total sequences, and this is consistent with the research results of Kelly et al. [47] and Pajarillo et al. [40]. In the cecum and colon, weaned piglets fed a diet containing 8% and 12% FBSPW increased the Firmicutes compared with the control, but the increase was not significant, indicating a favorable change. Firmicutes contain a large number of probiotics and fermentation bacteria [48,49], which may be related to the increase in dietary fiber content in the feed.

Our results showed that the relative abundances of *Streptococcus*, *Roseburia*, *Terrisporobacter*, *Prevotella*, *Solobacterium*, *Holdemanella*, *Actinobacillus*, *Blautia*, *Dorea*, *Butyricicoccus*, *Ruminiclostridium*, *E. xylanophilum*, *Oscillospira*, *Alloprevotella*, *Eubacterium*, *Clostridium*, and multiple unidentified genera were significantly different among groups, and compared with those in the control, the relative abundances of *Holdemanella*, *Butyricicoccus*, *Solobacterium*, *Clostridium*, and *Alloprevotella* in cecum microbiota of the piglets fed the basal diet with 12% FBSPW were significantly enriched, but that of *Oscillospira* was significantly reduced (*p* < 0.05; Appendix A). Most of these enriched genera can degrade cellulose and produce short-chain fatty acids [50,51,52,53,54,55,56], and short-chain fatty acids can be absorbed and utilized by the colon and cecum as energy substances and immunomodulatory factors [56,57,58,59,60]. For instance, Hirano et al. [61] reported that the enzymatic diversity of the *Clostridium thermocellum* cellulosome is crucial for the degradation of crystalline cellulose and plant biomass. Moreover, Lü et al. [62] found that the cellulose-degrading activity of cellulolytic *C. thermocellum* CTL-6 can be enhanced through co-culturing with a non-cellulolytic *Geobacillus* sp. W2-10. As a butyrate producer with probiotic potential, *Butyricicoccus pullicaecorum* is intrinsically tolerant to the stomach and small intestine conditions [51]. These results infer that the addition of 12% FBSPW to feed could improve the GM composition of weaned piglets. However, these significantly different cellulose-degrading genera did not correlate with serum parameters of the weaned piglets (Figure 6). These results indicated that the cecum microorganisms closely related to the feed compositions and those closely related to the serum parameters of weaned piglets were not consistent, which suggested that there is a bacterial interaction mechanism in cecum microbiota, although this mechanism needs further experiments to clarify.

## 5. Conclusions

The addition of 12% FBSPW to feed regulated the protein and lipid metabolisms of weaned piglets and the composition and function of GM. Notably, the addition of 12% FBSPW to feed increased the abundance of cellulose-degrading bacteria. Therefore, FBSPW is expected to become a new feed additive for weaned piglets.

## Figures and Tables

**Figure 1 animals-12-02728-f001:**
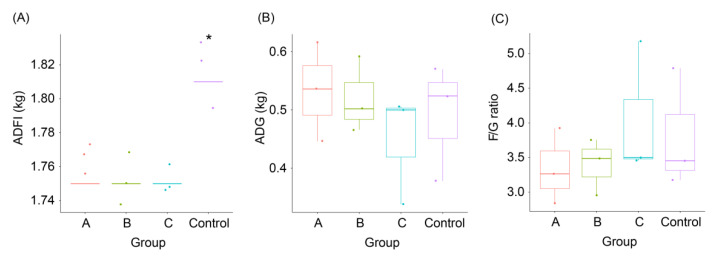
Growth performance of weaned piglets fed different feeds containing different fermented bamboo shoot processing wastes (FBSPW). (**A**) average daily feed intake (ADFI); (**B**) average daily gain (ADG); (**C**) feed/gain (F/G) ratio. The initial and final ages of the piglets were 38 and 89 days, respectively. Control, A, B, and C were the piglets fed a basal diet, a basal diet with 4% FBSPW, a basal diet with 8% FBSPW, and basal diet with 12% FBSPW, respectively. Dots in the boxplots indicate values measured in samples. * *p* < 0.05.

**Figure 2 animals-12-02728-f002:**
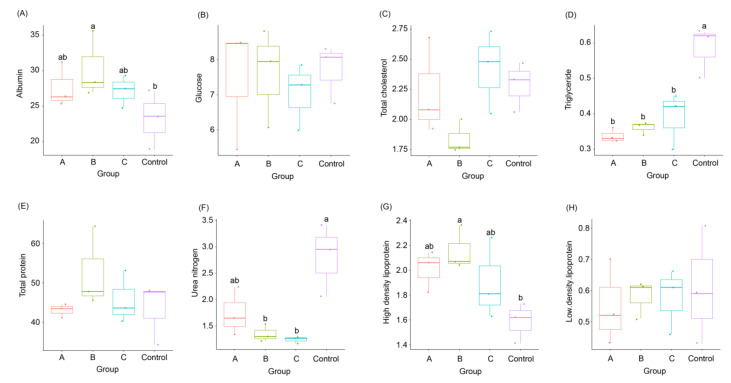
Serum biochemical indices of weaned piglets fed different feeds containing different fermented bamboo shoot processing wastes (FBSPW). (**A**) albumin; (**B**) glucose; (**C**) total cholesterol; (**D**) triglyceride; (**E**) total protein; (**F**) urea nitrogen; (**G**) high-density lipoprotein; (**H**) low-density lipoprotein. Control, A, B, and C were the piglets fed a basal diet, a basal diet with 4% FBSPW, a basal diet with 8% FBSPW, and a basal diet with 12% FBSPW, respectively. Dots in the boxplots indicate values measured in samples. Different letters above the boxes indicate significant differences (*p* < 0.05).

**Figure 3 animals-12-02728-f003:**
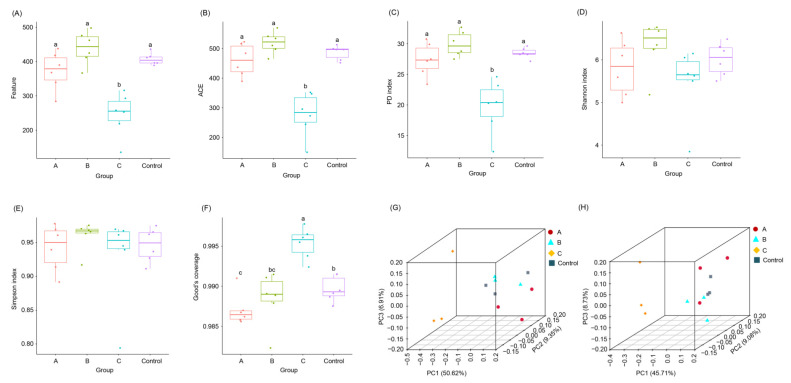
Boxplots of species features (**A**), ACE (**B**), PD (**C**), Shannon (**D**), Simpson (**E**), and Good’s coverage (**F**) indices, and PCoA profiles of the cecum (**G**) and colon (**H**) microbiota of weaned piglets fed different feeds containing different fermented bamboo shoot processing wastes (FBSPW). Control, A, B, and C were the piglets fed a basal diet, a basal diet with 4% FBSPW, a basal diet with 8% FBSPW, and a basal diet with 12% FBSPW, respectively. Dots in the boxplots indicate values measured in samples. Different letters above the boxes indicate significant differences (*p* < 0.05).

**Figure 4 animals-12-02728-f004:**
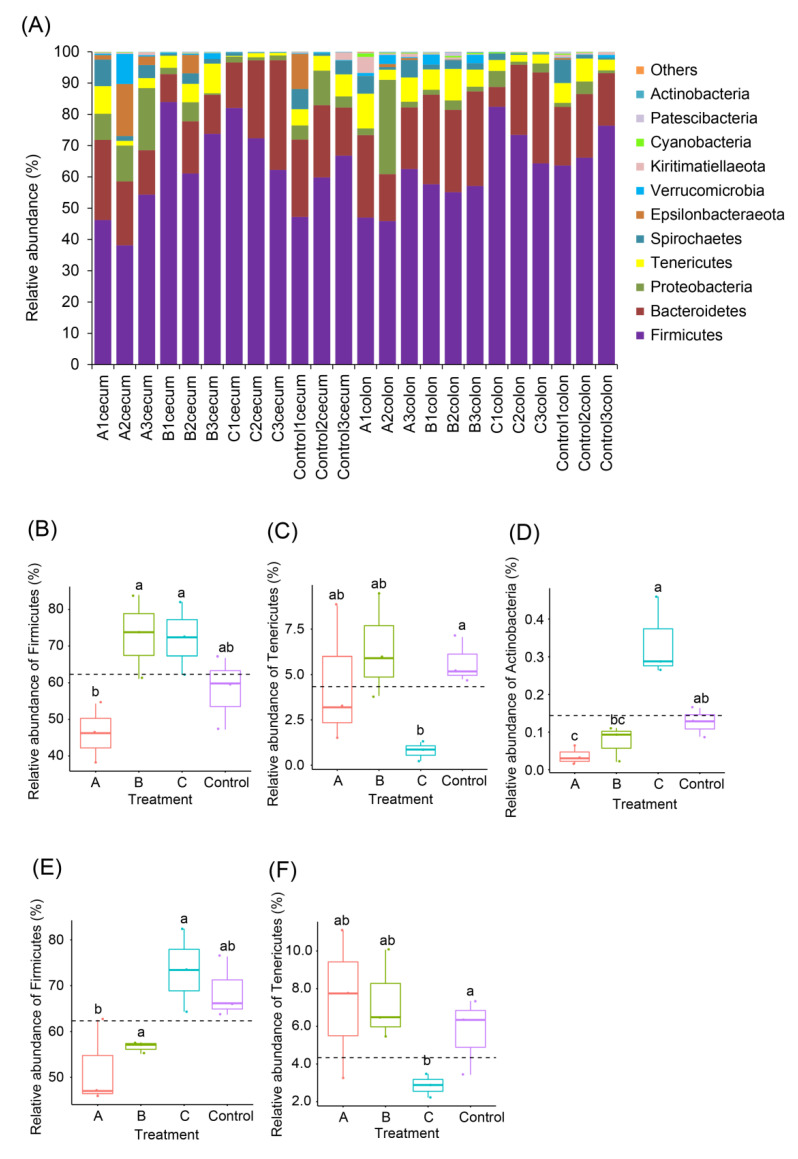
Relative abundances of phyla in the cecum and colon microbiota of weaned piglets fed different feeds containing different fermented bamboo shoot processing wastes (FBSPW). (**A**) Relative abundances of the dominant phyla. (**B**–**D**) A significant difference in the relative abundance of Firmicutes, Tenericutes, and Actinobacteria in cecum microbiota between different treatments, respectively. (**E**,**F**) A significant difference in the relative abundance of Firmicutes, and Tenericutes in colon microbiota between different treatments, respectively. Control, A, B, and C were the piglets fed a basal diet, a basal diet with 4% FBSPW, a basal diet with 8% FBSPW, and a basal diet with 12% FBSPW, respectively. Dots in the boxplots indicate values measured in samples. Different letters above the boxes indicate significant differences (*p* < 0.05).

**Figure 5 animals-12-02728-f005:**
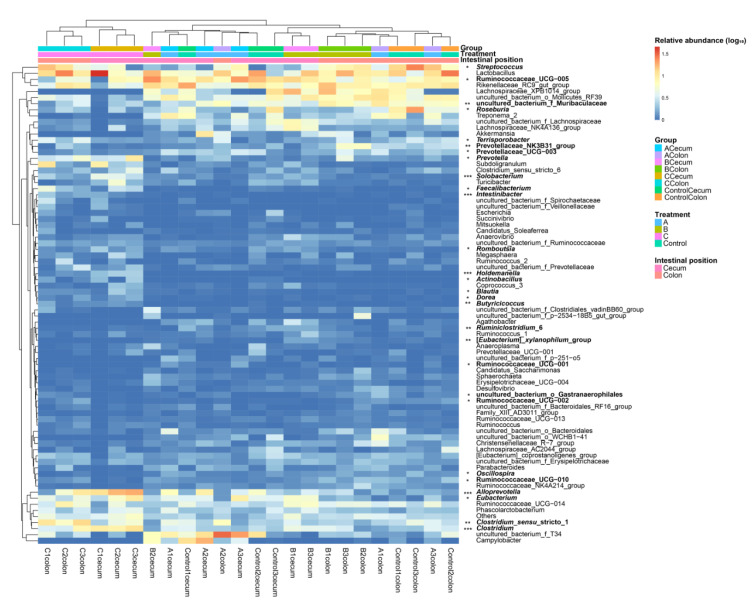
Heatmap profile shows the relative abundance of dominant genera in cecum and colon microbiota of weaned piglets fed different feeds containing different fermented bamboo shoot processing wastes (FBSPW). Control, A, B, and C were the piglets fed basal diet, basal diet with 4% FBSPW, basal diet with 8% FBSPW, and basal diet with 12% FBSPW, respectively. * *p* < 0.05; ** *p* < 0.01; *** *p* < 0.001.

**Figure 6 animals-12-02728-f006:**
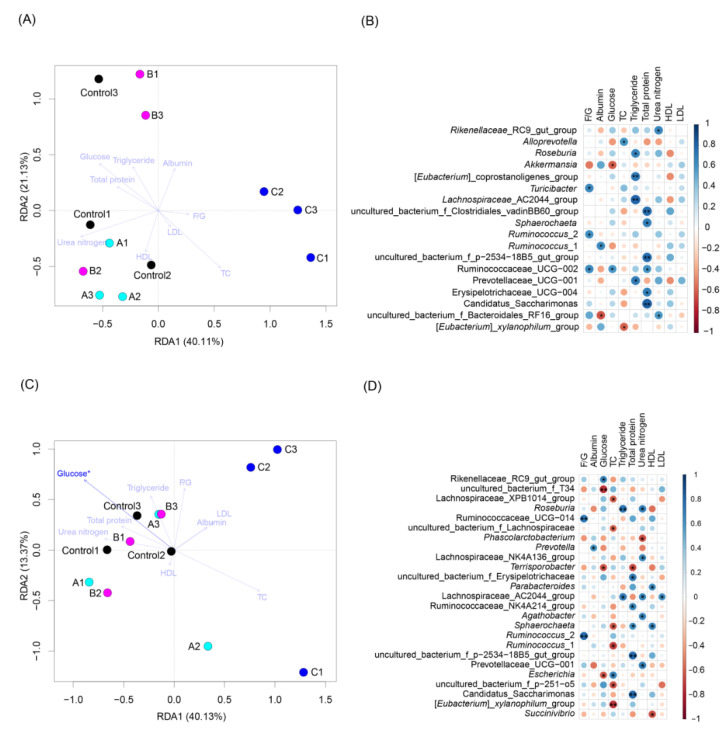
RDA profiles and bubble charts of cecum and colon microbiotas of weaned piglets fed different feeds containing different fermented bamboo shoot processing wastes (FBSPW). (**A**) RDA profile of cecum microbiota; (**B**) bubble chart of cecum microbiota; (**C**) RDA profile of colon microbiota; (**D**) bubble chart of colon microbiota. Control, A, B, and C were the piglets fed basal diet, basal diet with 4% FBSPW, basal diet with 8% FBSPW, and basal diet with 12% FBSPW, respectively. TC, total cholesterol; HDL, high density lipoprotein; LDL, low density lipoprotein. * *p* < 0.05; ** *p* < 0.01.

**Table 1 animals-12-02728-t001:** Composition and nutrient levels of diets (air-dry basis) used in the study. A, B, and C were the piglets fed a basal diet with 4% FBSPW, a basal diet with 8% FBSPW, and basal diet with 12% FBSPW, respectively. ADF, acid detergent fiber; NDF, neutral detergent fiber; FBSPW, fermented bamboo shoot processing waste; ME, metabolizable energy.

Items	Basal Diet	A	B	C
Ingredients (%)				
FBSPW	0	4	8	12
Corn	65	63	61	59
Soybean meal	20	19	18	17
Wheat bran	8	7	6	5
Calcium hydroxide	3	3	3	3
Salt	0.35	0.35	0.35	0.35
Soybean oil	2	2	2	2
Premix ^1^	1.65	1.65	1.65	1.65
Total	100	100	100	100
Calculated composition				
ME (MJ/kg)	13.79	13.62	13.47	13.30
Crude protein (%)	19.22	19.43	19.66	19.87
Crude fiber (%)	5.1	5.4	5.7	6.0
ADF (%)	7.62	8.00	8.37	8.72
NDF (%)	16.01	16.84	17.58	18.33
Lysine (%)	1.07	1.09	1.11	1.13
Met (%)	0.32	0.33	0.34	0.35
Calcium (%)	0.77	0.76	0.75	0.74
Total phosphorus (%)	0.68	0.66	0.65	0.64

^1^ Supplied the following (mg/kg): 20 mg/kg of Fe; 10 mg/kg of Cu; 100 mg/kg of Zn; 0.3 mg/kg of Se; 60 mg/kg of Ca; 10,000 IU/kg of vitamin A; 70 mg/kg of vitamin E; 1500 IU/kg of vitamin D3; 0.8 mg/kg of vitamin K; 2 mg/kg of vitamin B1; 4 mg/kg of vitamin B2; 5 mg/kg of Vitamin B6.

## Data Availability

All DNA sequences were deposited into the NCBI Sequence Read Archive database with accession number PRJNA827709.

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
