# Peer review of "Effects of Fermented Bamboo Shoot Processing Waste on Growth Performance, Serum Parameters, and Gut Microbiota of Weaned Piglets"

_animals, 2022, doi:10.3390/ani12202728_

Round 1

Reviewer 1 Report

The paper entitled “Effects of fermented bamboo shoot processing waste on growth performance, serum parameters, and gut microbiota of weaned piglets” deals with an interesting topic as is the use of by-products for a sustainable production. In general it is correctly and clearly written. However, there are some major points that need to be solved:

1- The information about the diets is not enough. There is no information about the ingredient that is replaced by the FBSPW. Table 1 only shows some calculated nutrients but some are missing, especially the fibre content, but also some AAs. And it would be better to show the analyzed composition (or both).

2- When explaining the data analysis, authors wrote that a One-way ANOVA was used (parametric) but later on results it is written that Kruskal-Wallis test was used (non-parametric). Please, clarify that. Were the data normally or non-normally distributed?

3- The authors show many microbiological results but the discussion of them is quite poor. This should be improved to better understand the importance of all the measurements. Also, the citation of more studies and the discussion of the agreements or disagreements with them would help to validate the results.  

4- It is discussed often that the cellulose content of the feed would affect different parameters, but the cellulose content is not given anywhere in the manuscript.

OTHER COMMENTS

- L81: “the fermentation seed solution was more than 10Ë„8 CFU/g.” Please, be more specific, because “more than” could mean 2x or 100x.

- L98: what is “normal immunization procedures”? It can change among countries so, please, specify.

- Table 1: a) energy is not a nutrient so change it to, for example, “Calculated composition”. b) No inorganic phosphorus or calcium sources included? c) P level seems a bit too high.

- L118: please, explain more the “overdose of anesthetics”.

- A brief explanation of the diversity indices would be very useful.

- L 274: …”and antinutritional factors”. Please, give examples.

Author Response

Responses to the comments

Comment

1- The information about the diets is not enough. There is no information about the ingredient that is replaced by the FBSPW. Table 1 only shows some calculated nutrients but some are missing, especially the fibre content, but also some AAs. And it would be better to show the analyzed composition (or both).

Response

Thank you very much for your comment. We have supplemented the contents of Table 1 according to the comment.

Comment

2- When explaining the data analysis, authors wrote that a One-way ANOVA was used (parametric) but later on results it is written that Kruskal-Wallis test was used (non-parametric). Please, clarify that. Were the data normally or non-normally distributed?

Response

Shapiro-Wilk normality test and Bartlett test of homogeneity of variances were conducted using R 4.0.3. When the data are normally distributed and the variances are homogeneous, one-way ANOVA was conducted using SPSS 25.0R 4.0.3. Duncan’s test was used for multiple comparisons of results with significant difference. Otherwise, Kruskal-Wallis rank sum test with Dunn post hoc tests were conducted using R 4.0.3 with FSA package. We have added the description in the Materials and Methods section.

Comment

3- The authors show many microbiological results but the discussion of them is quite poor. This should be improved to better understand the importance of all the measurements. Also, the citation of more studies and the discussion of the agreements or disagreements with them would help to validate the results.  

Response

Thank you very much for your comment. We have added some discussion according to your comment. However, since many results are statistical results of correlation, and there was no further experimental verification, we did not further discuss them. Although our results can provide other researchers with ideas for follow-up research, we’re worried that unduly discussion may mislead other researchers.

Comment

4- It is discussed often that the cellulose content of the feed would affect different parameters, but the cellulose content is not given anywhere in the manuscript.

Response

Thank you very much for your comment. We have added crude fiber contents of the diets in Table 1.

Comment

- L81: “the fermentation seed solution was more than 10Ë„8 CFU/g.” Please, be more specific, because “more than” could mean 2x or 100x.

Response

The total biomass in the fermentation seed solution was 2.4 ×109 CFU/g. We have revised the description. Thank you very much for your comment.

Comment

- L98: what is “normal immunization procedures”? It can change among countries so, please, specify.

Response

The piglets were vaccinated with classical swine fever vaccine at 1 - 2 h before colostrum, attenuated pseudorabies vaccine at age of 3 d, inactivated asthmatic vaccine at age of 10 d, swine streptococcosis vaccine at age of 17 d, attenuated paratyphus vaccine for piglets at age of 25 d, inactivated asthmatic vaccine at age of 30 d, and classical swine fever vaccine at age of 60 d. We have added the description in our revised manuscript. Thank you for your comment.

Comment

- Table 1: a) energy is not a nutrient so change it to, for example, “Calculated composition”. b) No inorganic phosphorus or calcium sources included? c) P level seems a bit too high.

Response

We have changed “Calculated nutrients” to “Calculated composition” according to your comment. The premix contains 60 mg/kg of calcium, and some other raw materials also contain calcium and phosphorus. The content of total phosphorus and calcium in the feed is indeed on the high side, which is due to the ratio made by the breeding company in order to make piglets grow faster and improve economic benefits. However, the calcium phosphorus ratio is between 1.1 and 1.2:1, which can ensure the normal absorption and utilization of piglets.

Comment

- L118: please, explain more the “overdose of anesthetics”.

Response

According to the national standard GB/T 39769-2021 of the People’s Republic of China, the piglets raised in the same fence were driven into a euthanasia box composed of transparent and closed hard plastic materials, and carbon dioxide was pumped into the box with high-pressure bottles of carbon dioxide with pressure gauges until the piglets stop breathing. We have added the national standard to our revised manuscript. Thank you for your comment.

Comment

- A brief explanation of the diversity indices would be very useful.

Response

We have added a brief explanation of the diversity indices in the Materials and Methods section. Thank you for your comment.

Comment

- L 274: …”and antinutritional factors”. Please, give examples.

Response

Thank you for your comment. We have added the examples according to your comment.

Reviewer 2 Report

Dear authors, the article has scientific merit, but some fundamental things need to be elucidated.

1 - What is the chemical composition of the product you are using? The fiber level and profile will have an impact on the microbiota, what would those levels of your treatment be?

Table 1 needs to present the nutritional levels of all treatments, not just the control. It is not possible that 12% inclusion will not affect the chemical composition of the rations. Describe the chemical composition more fully.

2 - What age at weaning of piglets? Why were they given a pre-trial diet for 7 days? (line 95). At what age did piglets start to consume the diet? Certainly the age of weaning influences the response of both performance and the microbiome.

3 - Was blood collected from only 3 animals per treatment? in line 110, the authors report that it was collected from 3 animals per group, but it must be corrected, because it is per treatment. What is the standard error of the mean for these samples, the n of 3 repetitions per treatment is too low, the deviation from the mean must have been too high.

4 - I consider an n 3 animals too low for the microbiome analysis, the variability between the microbiota is high. As the cost of analysis today is not that high, it does not justify such a low n. I believe the authors might call it an exploratory study.

5 - Why use a non-parametric test, Kruskal-Wallis, for performance data? I don't see any sense, because they are quantitative data. I also think that the graphical form is not the best way to present the performance, a table would be more appropriate.

6 - Figure 1 - the description must contain the initial and final age

7 - Figure 4 - Graphs B and C were confused and very bad to see the difference between the data. I suggest separating and presenting only the static differences.

8 - You start the discussion by reporting that the product has high levels of protein, in the introduction you put it as 131.4 g/kg, I do not consider this level as high. Protein content is certainly not the objective of using this product in the feed.

9 - There is no concise discussion about the effects of treatments on the microbiota, for example, in figure 6 the authors bring a correlation with bacteria with F/G, glucose triglyceride, among others. What is the impact of this for the animal?

Author Response

Comment

1 - What is the chemical composition of the product you are using? The fiber level and profile will have an impact on the microbiota, what would those levels of your treatment be?

Table 1 needs to present the nutritional levels of all treatments, not just the control. It is not possible that 12% inclusion will not affect the chemical composition of the rations. Describe the chemical composition more fully.

Response

Thank you very much for your comments. We have supplemented the contents of Table 1 according to the comment.

Comment

2 - What age at weaning of piglets? Why were they given a pre-trial diet for 7 days? (line 95). At what age did piglets start to consume the diet? Certainly the age of weaning influences the response of both performance and the microbiome.

Response

Thank you very much for your comments. The piglets were weaned at 25 days of age and purchased to the breeding company at 31 days of age. The piglets were pre-fed with basal diet for 7 days to alleviate the stress response of the piglets due to changes in living environment and transportation process. The piglets began to eat the experiment diet at the age of 38 days.

Comment

3 - Was blood collected from only 3 animals per treatment? in line 110, the authors report that it was collected from 3 animals per group, but it must be corrected, because it is per treatment. What is the standard error of the mean for these samples, the n of 3 repetitions per treatment is too low, the deviation from the mean must have been too high.

Response

Yes, the blood was collected from only 3 animals per treatment. We have changed the “group” to “treatment”. The standard errors of the means for these samples were showed in Table S1. We acknowledge that a small sample size causes the deviation from the mean to be on the high side, thus reducing the detection efficiency of difference significance to a certain extent, but this does not affect the reliability of the results that have detected significant differences. Therefore, our results are still valid, even though some potential differences were not detected because of the small sample size.

Comment

4 - I consider an n 3 animals too low for the microbiome analysis, the variability between the microbiota is high. As the cost of analysis today is not that high, it does not justify such a low n. I believe the authors might call it an exploratory study.

Response

We fully agree with your comment that the results would be more accurate if the sample size is increased. However, our funding is insufficient. Although the cost of sequencing today is not that high, the cost of purchasing piglets was still high. Although seventy-two piglets were purchased and used for experiment, we need to pay extra compensation for the piglets euthanized at the end of the experiment. Moreover, since a large number of previous studies have adopted the experimental design of three replicates for each treatment, it is difficult for each group to induce more replicates to pass the review of the Animal Ethics Committee. As we said earlier, small sample size does not affect the reliability of the results that have detected significant differences. Therefore, our results are still valid, even though some potential differences were not detected because of the small sample size.

Comment

5 - Why use a non-parametric test, Kruskal-Wallis, for performance data? I don't see any sense, because they are quantitative data. I also think that the graphical form is not the best way to present the performance, a table would be more appropriate.

Response

Shapiro-Wilk normality test and Bartlett test of homogeneity of variances were conducted using R 4.0.3. When the data are normally distributed and the variances are homogeneous, one-way ANOVA was conducted using SPSS 25.0R 4.0.3. Duncan’s test was used for multiple comparisons of results with significant difference. Otherwise, Kruskal-Wallis rank sum test with Dunn post hoc tests were conducted using R 4.0.3 with FSA package. We have added Table S1 to show the serum biochemical indices of weaned piglets.

Comment

6 - Figure 1 - the description must contain the initial and final age

Response

Thank you for your comments. We have added the initial and final ages in the figure legends.

Comment

7 - Figure 4 - Graphs B and C were confused and very bad to see the difference between the data. I suggest separating and presenting only the static differences.

Response

Thank you for your comment. We have revised the figure according to your comment and reanalyzed the data.

Comment

8 - You start the discussion by reporting that the product has high levels of protein, in the introduction you put it as 131.4 g/kg, I do not consider this level as high. Protein content is certainly not the objective of using this product in the feed.

Response

Thank you for your comment. In the start of Discussion section, we said that BSPW has high crude protein content, rather than adding it to the feed can increase the protein content of the feed. Although protein content is certainly not the objective of using this product in the feed, as a feed additive, the protein content of FBSPW is still an important characteristic.

Comment

9 - There is no concise discussion about the effects of treatments on the microbiota, for example, in figure 6 the authors bring a correlation with bacteria with F/G, glucose triglyceride, among others. What is the impact of this for the animal?

Response

Thank you very much for your comment. We have added some discussion according to your comment. However, since many results are just statistical results of correlation, and there was no further experimental verification, we did not further discuss them. Although our results can provide other researchers with ideas for follow-up research, we’re worried that unduly discussion may mislead other researchers.

Round 2

Reviewer 1 Report

I would like to thank the authors for accepting and answering my comments/questions. However I still have some comments:

- The composition of the diet is still not complete for me. In my opinion, when testing fibrous materials, analyzing CF is not enough. At least, the composition of the tested ingredient with the different fractions of fiber should be given. On the other hand, regarding the phosphorus content, I still doubt that it can only come from the ingredients used without the addition of any inorganic source.

- L161: change "are" for "were"

- L179: change "are" for "were"

- Fig. 4. The addition in the headline does not contribute to give more information. I think it was clear enough with the initial sentence. I would, therefore, remove it and leave it as it was in the previous version.

- L397: correct "crucial" (it is written as "cruial")

Author Response

Comment

- The composition of the diet is still not complete for me. In my opinion, when testing fibrous materials, analyzing CF is not enough. At least, the composition of the tested ingredient with the different fractions of fiber should be given. On the other hand, regarding the phosphorus content, I still doubt that it can only come from the ingredients used without the addition of any inorganic source.

Responses

Thank you very much for your comment. We have added the ADF and NDF contents in the Table 1. The premix contains a certain amount of inorganic phosphorus, but since this content is considered as a trade secret, we do not know the content of inorganic phosphorus in the premix.

Comment

- L161: change "are" for "were"

Response

Thank you for your comment. We have changed the word according to your comment.

Comment

- L179: change "are" for "were"

Response

Thank you for your comment. We have changed the word according to your comment.

Comment

- Fig. 4. The addition in the headline does not contribute to give more information. I think it was clear enough with the initial sentence. I would, therefore, remove it and leave it as it was in the previous version.

Response

Thank you very much for your comment. If the addition in the headline is removed, we cannot determine whether the dominant phyla in (B) - (F) from cecum or colon microbiota. Therefore, we have only condensed these sentences, but not deleted them completely.

Comment

- L397: correct "crucial" (it is written as "cruial")

Response

Thank you for your comment. We have revised the word according to your comment.

Reviewer 2 Report

The authors answered my doubts